# Detection of a 4 bp Mutation in the 3′UTR Region of Goat *Sox9* Gene and Its Effect on the Growth Traits

**DOI:** 10.3390/ani10040672

**Published:** 2020-04-13

**Authors:** Libang He, Yi Bi, Ruolan Wang, Chuanying Pan, Hong Chen, Xianyong Lan, Lei Qu

**Affiliations:** 1College of Animal Science and Technology, Northwest A&F University, Yangling 712100, Shaanxi, China; helibangyl@163.com (L.H.);; 2Shaanxi Key Laboratory of Molecular Biology for Agriculture, Yangling 712100, Shaanxi, China; 3Shaanxi Provincial Engineering and Technology Research Center of Cashmere Goats, Yulin University, Yulin 719000, Shaanxi, China; 4Life Science Research Center, Yulin University, Yulin 719000, Shaanxi, China

**Keywords:** growth performance, chondrocyte, *Sox9*, polymorphism, goat

## Abstract

**Simple Summary:**

The sex determining region Y (SRY)-type high mobility group (HMG) box 9 (*Sox9*) gene is critically important in the formation and development of cartilage and is considered the “main regulator” of chondrogenesis. Additionally, a large number of studies have shown that mutations in a single allele of human *Sox9* can lead to campomelic dysplasia syndrome. Therefore, the mutations of *Sox9* have been the subject of increasing interest among researchers. However, no studies to date have examined the association between *Sox9* gene variants and growth traits in goats. Here, we detected a 4 bp indel in the 3′Untranslated Regions (3′UTR) region of *Sox9* in Shaanbei white cashmere (SBWC) goats (*n* = 1109) and studied the association between this indel and growth traits. The 4 bp indel of *Sox9* was significantly associated with body length, heart girth, hip width, and all body measurement indexes (*p* < 0.05) in SBWC goats. Thus, this deletion could be used as an effective molecular marker for maximizing the growth traits of goats in breeding programs.

**Abstract:**

The SRY-type HMG box 9 (*Sox9*) gene plays an important role in chondrocyte development as well as changes in hypertrophic chondrocytes, indicating that *Sox9* can regulate growth in animals. However, no studies to date have examined the correlation between variations in *Sox9* and growth traits in goats. Here, we found a 4 bp indel in the 3′UTR of *Sox9* and verified its association with growth traits in Shaanbei white cashmere goats (*n* = 1109). The frequencies of two genotypes (ID and II) were 0.397 and 0.603, respectively, and polymorphic information content (PIC) values showed that the indel had a medium PIC (PIC > 0.25). The 4 bp indel was significantly correlated with body length (*p* = 0.006), heart girth (*p* = 0.001), and hip width (*p* = 4.37 × 10 ^−4^). Notably, individuals with the ID genotype had significantly superior phenotypic traits compared with individuals bearing the II genotype. Hence, we speculated that the 4 bp indel is an important mutation affecting growth traits in goat, and may serve as an effective DNA molecular marker for marker-assisted selection in goat breeding programs.

## 1. Introduction

The sex determining region Y (SRY)-type high mobility group (HMG) box 9 (*Sox9*) gene is an important member of the SRY-type HMG box (*Sox*) gene family. It plays an essential role in cell differentiation in multiple tissues during embryonic development and in adult [1]. As a transcription factor, *Sox9* gene plays a pivotal role in mammalian sex determination during embryonic development [2]. In addition, *Sox9* is crucial for regulating reproduction. Previously, we have shown that the expression of *Sox9* was significantly associated with pig reproduction traits, and that it plays a critical role in testes development [3].

Meanwhile, *Sox9* has also been documented as playing a role in chondrocyte reproduction, differentiation, and the cartilage-specific extracellular matrix, thereby facilitating chondrogenesis [4,5]. The molecular mechanism underlying the regulation of chondrogenesis involves the binding of the HMG box of *Sox9* to a specific sequence on the DNA groove [6]. Next, the DNA strand is bent, and the double helix structure is unwound, leading to the transcription of the target gene [7,8]. Additionally, previous research has confirmed protein encoded by *Sox9*, which is a powerful activator of transcription both in vivo and in vitro, can bind intron 1 of *Col2al* (II collagen gene) at a special site and directly regulate the expression of this gene, which is specifically expressed in cartilage cells [9,10,11]. In addition, the differentiation of undifferentiated mesenchymal stem cells (MSCs) and polymerization of mesenchymal cells require the presence of *Sox9* in the early stages of chondrogenesis. Chondrocyte and MSCs can develop into the body’s skeleton during the late development of individuals, and *Sox9* plays an important role in this process [12]. Therefore, we hypothesize that *Sox9* is a candidate gene that could affect growth trait.

The goat industry, which is one of the most ancient and productive livestock industries, plays an important role in the Chinese economy. However, the current state of goat breeding in China, in particular its low efficiency and the imbalance between demand and supply, has become a major problem requiring immediate attention [13,14,15,16]. In northern Shaanxi, China, Shaanbei white cashmere (SBWC) goats are a crucial breed used for wool and meat, and are cold-tolerant, highly adaptable, and feedstuff-tolerant. However, SBWC goats still have the problem of poor growth traits in their actual production [17]. Thus, the way in which to improve the production performance of goats needs to be solved urgently [18]. There are many factors can affect growth traits of goats, among which the genetic factor plays a key regulatory role [19]. The main genetic variations are insertion/deletions (indels), SNPs, and CNVs, among others. Among them, insertion/deletions (indels) are easily identified by simple PCR amplification and agarose gel electrophoresis [20], and extensively exist in eukaryotic genomes. Indels were used in marker-assisted selection (MAS), forming a convenient and efficient method for breeding selection that is not affected by the environment [21]. From the previous studies, we found that indels of a gene are closely related to certain traits [22], including the growth trait of cashmere goats [23], such as *Glutaminyl-peptide cyclotransferase-like* (*QPCTL*) [24], *Cell division cycle 25A* (*CDC25A*) [25], and *Cyclin-dependent kinase inhibitor 3* (*CDKN3*) [26]. 

However, few studies have examined the association between *Sox9* and growth traits in SBWC goats. Herein, we aimed to identify indels in the *Sox9* that could enhance the process of goat-selective breeding. To increase the reliability of this experiment and characterize the impact of the 4 bp indel on the growth traits, 1109 individuals of SBWC goats were analyzed. These findings could identify molecular markers for MAS programs that could be used to improve the growth traits of local breeds in the goat industry.

## 2. Materials and Methods 

All experiments involving animals were approved by the Faculty Animal Policy and Welfare Committee of Northwest A&F University (protocol number NWAFAC1008). Furthermore, the care and use of experimental animals was completely in accord with the local animal welfare laws and policies.

### 2.1. DNA Samples and Data Collection

All tested female goats (12 to 18 months of age) were just reaching physical maturity when they were selected randomly (*n* = 1109). All goats in the SBWC Breeding Farm received the same diet and were reared under the same set of standard conditions after weaning in Yulin, Shaanxi, China. Ear tissue samples were collected for subsequent DNA analyses. Data on SBWC growth-related traits were collected by staff in farms, including body height (BH), height across the hip (HH), body length (BL), heart girth (HG), Cannon circumference (CC), chest depth (CD), chest width (CW), and hip width (HW) [27,28].

### 2.2. DNA Isolation and DNA Pool Construction

Genomic DNA was extracted from the goat ear tissues using a high-salt extraction protocol [29,30]. The extracted DNA samples were quantified by Nanodrop 1000 (Thermo Fisher Scientific Inc., Wilmington, DE, USA), and then diluted to 20 ng/μL and stored at −20 °C [19,23]. Next, a DNA pool of 50 DNA samples was constructed. The 50 DNA samples were selected randomly from the experimental samples to explore the variation in the goat *Sox9* gene. Then, according to the polymorphism of this result, we could judge whether to continue to expand the sample at this site for analysis.

### 2.3. Primer Design and Genotyping

On the basis of the Ensembl indel database (http://www.ensembl.org/index.html), four potential indel sites were detected in the goat *Sox9* gene, which were used to design four primers with the software Primer Premier 5 to test the indel of *Sox9* (Table 1). Subsequently, PCR-based agarose gel electrophoresis amplification of fragment length polymorphism was used to score the genotype of the indel. Indel identification and genotyping were performed by using touch-down PCR in a 13 μL reaction mixture containing 50 ng of genomic DNA [28]. Each PCR product was electrophoresed on a 3.5% agarose gel stained with ethidium bromide to identify the indel locus for subsequent sequencing [19,31].

### 2.4. Statistical Analysis

Genotypic frequencies, allelic frequencies, and Hardy–Weinberg equilibrium (HWE) of the indel locus in *Sox9* were calculated using the SHEsis program (http://analysis.bio-x.cn/myAnalysis.php) [32]. Population indexes (heterozygosity, He; homozygosity, Ho; polymorphic information content, PIC) were calculated online using Nei’s methods (http://www.msrcall.com/Gdicall.aspx) [33]. The χ^2^ test was conducted to evaluate HWE using the SHEsis online platform (http://analysis.bio-x.cn/myAnalysis.php). According to the published linear model [23,31], the association between the indel locus and growth traits in SBWC goats was analyzed using a *t*-test in SPSS software (version 24.0) (International Business Machines Corporation, New York, NY, USA).

## 3. Results

### 3.1. Identification of a 3′UTR 4 bp Indel within Sox9

PCR was performed with the four synthesized primers using a mixed DNA pool of samples from SBWC goats (50 samples) to verify the indels in *Sox9* (Table 1). Ultimately, a novel 4 bp indel of the *Sox9* 3′UTR region (NC_030826.1: g.58090106-58090109delTCGC; rs649476917) was detected in SBWC goats using primer 2. Agarose gel electrophoresis of the PCR products and sequence diagrams of the novel indel showed that the 4 bp indel was polymorphic and had two genotypes (Figure 1 and Figure 2). Specifically, genotype II (homozygote) showed one band (166 bp), and genotype ID (heterozygote) exhibited two bands (166 and 162 bp), whereas the homozygous genotype DD (Deletion/Deletion) was not detected (Figure 1 and Figure 2). Meanwhile, the indel sequence that we detected was consistent with the sequence registered at the available Ensembl indel database.

### 3.2. Analysis of Genotype and Allele Frequencies

The genotype and allele frequencies of this polymorphism in SBWC goats (*n* = 1109) were calculated (Table 2). The frequencies of genotypes ID and II were 0.397 and 0.603, respectively. Subsequently, the population indices (Ho, He, Ne, and PIC) were calculated in the current locus on the basis of the frequency numbers of the different genotypes (Table 2). It showed that the numerical value of Ho was more than 0.500, and the PIC values were greater than 0.25, indicating that the indel had medium PIC. Additionally, the genotype distribution was not in HWE (*p* < 0.05) in totality.

### 3.3. Association Analysis between Indel Genotypes and Growth Traits in SBWC Goats

The body traits that we measured are a direct reflection of skeleton structure and are also related to the physiological function and the production performance of goats [15]. The associations between the 3′UTR 4 bp indel and growth-related traits were listed in the Table 3. We found that the 4 bp indel was extremely significantly related to body length (*p* = 0.006), heart girth (*p* = 0.001), and hip width (*p* = 4.37 × 10^−4^). These growth traits can thus be used to improve the production of livestock. Additionally, individuals with the ID genotype had superior values of all phenotypes relative to II individuals for all of the significantly associated characteristics. Surprisingly, all of the body measurement indexes were significantly correlated (*p* < 0.05); for example, the correlation between body length and chest circumference index was extremely significant (*p* < 0.01). Thus, this method can be used to analyze the body measurement of animals, which determines whether each body part is fully developed and whether body part is symmetrical and conforms to the characteristics of a certain production type and variety.

## 4. Discussion

In this paper, we detected a genetic polymorphism in a 4 bp indel located in the 3′UTR region among the four possible variants with the following two genotypes: II (homozygote insertion type) and ID (heterozygote type). Mutations in the *Sox9* can lead to campomelic dysplasia syndrome, a rare and lethal congenital skeletal dysplasia syndrome characterized by an endochondral osteogenesis disorder [1,34,35]. In addition, we hypothesized that this site is linked to other unknown genes; thus, this site might have been subjected to genetic drift or disease because of the linkage imbalance. This hypothesis would explain the apparent absence of the DD genotype. 

To further explore the correlation between the 4 bp indel and growth traits, we analyzed the association in SBWC goats (*n* = 1109). We found that the genotype and allele frequencies at this locus were not in HWE, which may be explained by migration, genetic drift, and artificial selection [36,37]. In addition, on the basis of the PIC value (0.25 < *p* < 0.50), this locus was in a medium polymorphism, indicating that it has a high potential value for selection in goat breeds [38,39]. The 4 bp indel was shown to be strongly correlated with growth traits (e.g., BL, HG, and HW) in SBWC goats. Therefore, we expected that this 4 bp indel could be used as a molecular maker to improve the reproduction and production performance in goats via MAS breeding. 

*Sox9* plays a critical role in chondrogenesis, bone formation, and the treatment for cartilage diseases [40]. This gene comprises an HMG-box (high mobility group-box) region that can specifically combine with the minor groove of DNA and ultimately modulate the expression of the relevant genes. For example, *Sox9* can initiate chondrogenic differentiation by regulating the expression of SRY-type HMG box5 (*Sox5*) and SRY-type HMG box6 (*Sox6*) [41,42]. Additionally, it is a key chondrogenic transcription factor, and the entire network of regulatory mechanisms depends on and affects *Sox9* expression and activity [43]. 

There are several possible mechanisms by which *Sox9* might inhibit chondrocyte hypertrophy. Studies have shown that *Sox9*, as a major transcriptional activator, maintains the chondrocyte through a phosphoinositide 3-kinase (PI3K) -AKT pathway [44,45], as well as preventing chondrocytes from becoming hypertrophic [12]. In addition, *Sox9* could block Wnt signaling through inducing the degradation of β-catenin to inhibit chondrogenic hypertrophy [46,47]. Moreover, *Sox9* also blocks the activity of Runt-related transcription factor 2 (Runx2) [48], which plays a crucial role in inducing chondrocyte maturation. 

Several previous studies have shown that 3′UTR variants can affect gene transcription [49,50], and structural changes in the 3′UTR region are closely related to livestock and poultry production performance [51,52,53]. The 3′UTR is generally thought to regulate gene expression by affecting the stability of mRNA. Changes in mRNA abundance are a key means by which post-transcriptional gene expression is regulated. It can cause changes in protein levels of related genes in the upstream and downstream, thereby affecting the expression of gene. Thus, the 4 bp indel of the 3′UTR region may play an important role in regulating the *Sox9* gene. Although the exact mechanism was not clear, association analysis of large samples (*n* = 1109) showed that the 4 bp indel of *Sox9* is closely related to the growth traits of goats. Hence, we speculated that the 4 bp indel might affect the expression process of *Sox9*, thereby resulting in the change in growth traits. 

*Sox9* is not only involved in the expression of genes in specific cartilage tissue, but also in the regulation of the expression of mesenchymal stem cells. Moreover, it plays an important role in regulating the hypertrophy and ossification of chondrocytes. Ultimately, we speculate that *Sox9* may mediate changes in growth traits in goats by affecting the growth of bone. Further study is required to determine whether this 4 bp indel would affect the expression of *Sox9* gene in different tissues and influences the binding of miRNA.

## 5. Conclusions

Briefly, the 4 bp indel of the *Sox9* gene in the 3′UTR region was proven to be strongly associated with growth traits (body length and heart girth), suggesting that this indel might be a potential DNA marker in goat MAS breeding in connection with production performance. These results provide a scientific basis for the development of growth traits and genetic resources in goat.

## Figures and Tables

**Figure 1 animals-10-00672-f001:**
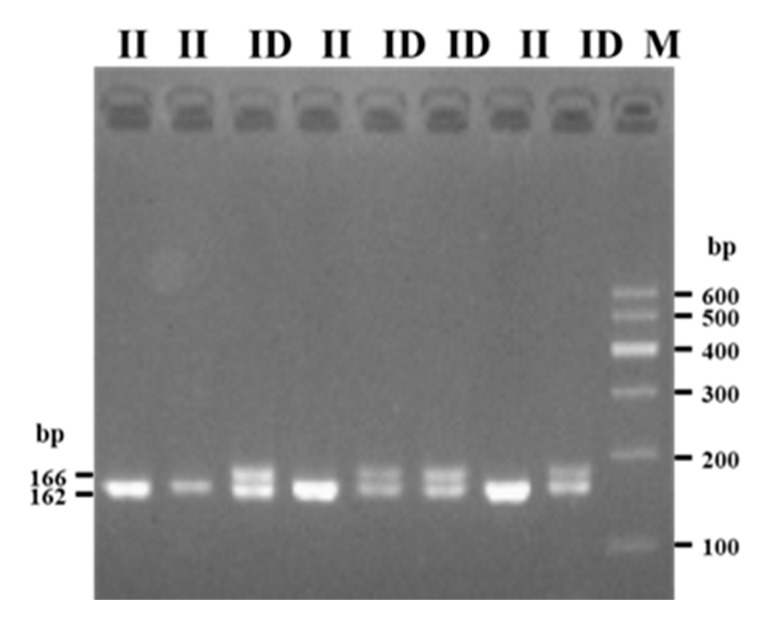
Agarose electrophoresis of goat *Sox9*-4 bp insertion/deletion (indel).

**Figure 2 animals-10-00672-f002:**
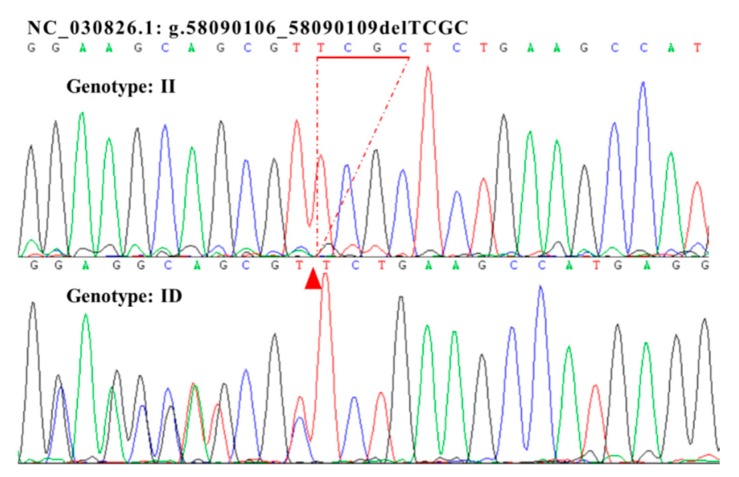
The sequencing chromas for the 4 bp indel in the goat *Sox9* gene. Sequencing chromas showed homozygotic insertion type (II) and heterozygote type (ID).

**Table 1 animals-10-00672-t001:** Primers for PCR amplification of SRY-type HMG box 9 (*Sox9*) gene.

Primers Name	Sequences (5′-3′)	Sizes (bp)	Function	Location	Note
P1	F: GCTATTCTTTGCCGCCCTGTG	112/108	Indel detection	3′UTR	Original design
	R: TCTCGGAGCAACTAAGCCTGTG				
P2	F: AGTGCCCTTTTCTCCTCCTA	166/162	Indel detection	3′UTR	Original design
	R: TGACCCTCCACTACCTCTTT				
P3	F: CCTACCACCACCATCTAAGTT	172/168	Indel detection	3′UTR	Original design
	R: CCCTTTCTGTTCCATACCAATA				
P4	F: TCCTTGCGGTCTCGGTGTTC	242/238	Indel detection	3′UTR	Original design
	R: AAGCCCAGAAACTGCCTTAACG				

F is the upstream primer, and R is the downstream primer.

**Table 2 animals-10-00672-t002:** Genetic parameters of the indel within *Sox9* in Shaanbei white cashmere goat.

Observed Genotypes (*N* = 1109)	Frequencies		Ho	He	PIC	χ^2^ (*p*-Value)
	Genotypes	Alleles				
DD (0)	0	0.198(D)	0.682	0.318	0.268	67.916 (*p* = 0.0001)
ID (440)	0.397	0.802(I)				
II (669)	0.603					

Note: Ho, homozygosity; He, heterozygosity; PIC, polymorphism information content.

**Table 3 animals-10-00672-t003:** Associations of the indel with growth traits in SBWC goats.

Traits	Genotypes (bp)		*p*-Values
	ID	II	
BH (cm)	54.78 ± 0.18 (*n* = 439)	55.01 ± 0.14 (*n* = 668)	0.329
BL (cm)	66.84 ^a^ ± 0.25 (*n* = 439)	65.92 ^c^ ± 0.22 (*n* = 668)	0.006
HH (cm)	57.77 ± 0.18 (*n* = 439)	57.87 ± 0.15 (*n* = 667)	0.708
CW (cm)	19.13 ± 0.14 (*n* = 439)	19.12 ± 0.11 (*n* = 668)	0.974
CD (cm)	27.67 ^a^ ± 0.14 (*n* = 439)	27.20 ^b^ ± 0.11 (*n* = 668)	0.010
HG (cm)	83.15 ^a^ ± 0.48 (*n* = 439)	81.13 ^c^ ± 0.41 (*n* = 668)	0.001
CC (cm)	8.05 ^a^ ± 0.03 (*n* = 440)	7.95 ^b^ ± 0.03 (*n* = 668)	0.041
HW (cm)	16.71 ^a^ ± 0.16 (*n* = 440)	15.92 ^c^ ± 0.14 (*n* = 668)	4.37 × 10^−4^
BTI	124.33 ^a^ ± 0.50 (*n* = 439)	122.80 ^b^ ± 0.50 (*n* = 668)	0.026
BLI	122.40 ^a^ ± 0.52 (*n* = 439)	120.10 ^c^ ± 0.40 (*n* = 668)	3.83 × 10^−4^
CCI	152.19 ^a^ ± 0.90 (*n* = 439)	147.48 ^c^ ± 0.73 (*n* = 668)	5.2 × 10^−5^
TCI	14.75 ^a^ ± 0.08 (*n* = 439)	14.46 ^c^ ± 0.06 (*n* = 668)	0.004
CWI	69.28 ^a^ ± 0.45 (*n* = 439)	70.49 ^b^ ± 0.34 (*n* = 668)	0.029
HWI	119.86 ^a^ ± 1.53 (*n* = 439)	125.56 ^b^ ± 1.16 (*n* = 668)	0.003

Note: BH: body height, BL: body length, HH: height at hip cross, CW: chest width, CD: chest depth, HG: heart girth, CC: cannon circumference, HW: hip width, BTI: body trunk index, BLI: body length index, CCI: chest circumference index, TCI: tube confining index, CWI: chest width index, HWI: hip width index. The mean values with different superscripts (^a, b^) within the same row differ significantly at the *p* < 0.05 level; (^a,^
^c^) within the same row extremely differ significantly at the *p* < 0.01 level. II = insertion genotype; ID = heterozygote genotype.

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
