# Peer review of "Detection of a 4 bp Mutation in the 3′UTR Region of Goat *Sox9* Gene and Its Effect on the Growth Traits"

_animals, 2020, doi:10.3390/ani10040672_

Round 1
Reviewer 1 Report
Reviews animals-750493
The manuscript describes the results of association between a mutation of Sox9 and growth performance in a large population. It would be helpful to selective breeding in SBWC population.
The text should be reviewed for scientific wording and style. Some few examples are given below:
Line 28
Line 74 while ……
Line 153 These traits can be used as symbols to improve the production of livestock.
Line 168 applying to MAS breeding
Line 204 by improve the growth traits.
Line 223
Line 236 and study has shown
Line 254 Therefore, it can be seen that Sox9 gene is not only involved in the expression of genes in specific cartilage tissue, but also regulates the expression of mesenchymal stem cells.
Line 265 These results provide scientific basis for the development of growth traits and genetic resources in goat.
Regarding the scientific content
pleased provide the physiological state of does given the strong effects of pregnancy on body weight and body conformation.
Since body weight (BW) has been collected, what are the effects of the mutation on it? Please offer the important information.
The animal model used in this study is too simple to detect the true effects of mutation. In general, body weight is treated as a covariate when analyzing the body conformation traits.
Line 155 “Even more surprising, all the body measurements index was significantly correlated, among them, the body length and Chest circumference index was extremely significant.” It would be better to give/reference the detail results.
Author Response
Dear Reviewer
I have submit a cover letter to response your comments, please see the attachment.
Kind regards,
Libang He

Reviewer 2 Report
Dear authors, the manuscript is interesting, but in order to be accepted for publication few corrections and explanations should be done.
The other indel in P1, P3 and P4 were not found in the population? How was the genotypes of the population for these regions?
According to the agarosis gel, it seems that the genotypes are ID (166bp and 162bp) and DD (162bp). The missing genotype is II (166bp). (Figure I)
The sequencing is also a bit confusing (Figure I). The named genotype II has the 4bp (TCGC) insertion and it corresponds to the 166bp fragment. The other genotype ID has the not corresponding picks (common in indel), but it happens before the indel. Is the sequencing being read in 5’?
In sequencing, the II is described as the “homozygous deletion”, shouldn’t it be the homozygous insertion”.
The genotypes should be better explained.
The title of 3.2 item should be replaced by “Genotype and allele frequencies” as well as the title of Table 2.
There was one missing genotype and large number of animals were tested. Do you think it is deleterious?
The title says that the indel is located in 3´UTR but in the primer description it is inserted in the promoter. What is the correct?
If the deletion is bad, how the heterozygous can have better performance than the homozygous?
Are there any other articles about associations of SOX9 polymorphism or expression with growth traits in cattle, sheep or pig? It should be used in the Discussion.
Author Response
Dear Reviewer
I have submitted a cover letter to response your comments, please see the attachment.
Kind regards, Libang He
Reviewer 3 Report
The authors present a paper on detection of a 4-bp mutation in the 3'UTR region of 2 goat Sox9 gene and its effect on the growth traits. The topic of the paper seems to be original and interesting even if there are some points of concern that make the manuscript not acceptable in the present form.
General comment: even if the English language is not bad, please have it reviewed by a native English speaker as there were many grammatical mistakes throughout the manuscript, especially word choices and prepositional usage.
Some aspects about the experimental design, in my opinion, are not understandable (i.e. number of samples used, details on the molecular tecnique used in lab, etc.).
The effect of Sox9 gene on livestock production needs to be better explained. It could be useful to do it, moving several parts from discussion to introduction section (i.e. claryfing the sentence in line 153).
More in detail:
-please check the first page about the respect of the editorial rules (email address). Also the references section needs to be revised.
-keywords: it could be better to avoid words already included in the title.
Line 39: Consider editing this sentence in “the frequencies of the two genotypes (ID and II) were […]”
-line 61, please use italic style for “in vivo” and “vitro”. Same for the gene acronyms in lines 62, 63, and 102.
Line 70: Consider editing this sentence in “[…] the Chinese economy”;
Lines 80 to 82: Please consider rephrasing this sentence, is not clear.
Line 82: Delete “n=1109”.
Material and methods
Line 91: I can hypotize that the authors used only females to avoid the sex effect. It could be better to specify this aspect. Moreover in how many farms did the authors collected the samples? Did the authors verify if the collected animales were related?
Line 93: The DNA samples were isolated from ear tissues. I hypothesized was made to test the expression of the Sox9 gene in the cartilaginous tissue. But then, reading the Results section, there were no traces of this test. So, why have the authors isolated the DNA from ear and not from blood? (as usual in these types of experiments);
Line 101: if I understood well the authors used for the molecular analysis only 50 samples (on more than 1000 collected): what is the reason? Moreover in line 151 the authors said “1109 individuals which were genotyped in this study”: after this sentence the reader can understand that all the samples were genotyped…. I am very confused about this aspect.
Line 116: please replace “polymorphism” with “polymorphic”. Same in line 143.
Line 123-130: again some aspects that need to be clarified…. The authors mentioned 4 couples of primers but only results about 1 couple are reported…. Please clarify this aspect.
Line 128-129 and Figure 1: The authors reported: ”In detail, the genotype II showed one band (166 bp), genotype ID exhibited two bands (166 and 162 bp) yet genotype DD was not detected (Fig. 1 and 2).”
From the reported figure, it seems like that the II genotype is represented by the single band of 162 bp (I allele 162 bp), while the D allele is that of 166 bp. The figure does not represent what the authors referred.
Typing mistakes: Line 139 and 145: “table 2”;
Line 157: “chest”
Lines 155-157: the sentence is not understandable. Moreover it could be better to specify a significance level (and not extremely significant).
Lines 160-168: this part is not adequate for results section and contains repetitions from introduction.
Discussion
A real discussion section is missing in the paper. In this section the authors need to discuss their findings with the relevant available literature. In the present version the majority of the sentences could be summarized and moved to the introduction section, avoiding repetitions.
Line 197 to 199: I suggest the authors to rephrase this sentence, to make it more understandable.
Line 213 to 215: I agree with the authors hypothesis. I would like to suggest to investigate if this is also linked with the different sexual animal lineages (this is also linked with the previous comment, ”why only female goats?”). There are proves of Sox9 DD genotype males?
The discussion about miRNA is totally speculative because not supported from results: I suggest to delete.
Line 264, please replace “production” with “productive”.
Author Response

(The authors gave the same response as above.)

Round 2
Reviewer 2 Report
Manuscript accepted for publication.
Author Response
Dear Reviewer
Thank you so much for your assistance.
Kind regards,
Libang He
Reviewer 3 Report
General comment:
I would like to thank the authors for their efforts in improving the manuscript that in the present revised version is now more understandable. However, I still have few additional comments.
[Response 3]
Deeply Grateful for your advice. The discussion has been rewritten. Several parts from discussion have been remove to introduction section.
Lines 183-196: this part should be still summarized.
[Response 7]
Thanks. These parts have been modified.
Line 54, it could be better to write “in vivo and in vitro”
[Response 11] “[…] In addition, this batch of data came from different goat farms in the Shaanbei white cashmere goat base of Yulin, the collected animals were not related.”
Line 88-89: In the Manuscript is reported as follows: “The tested 1,109 female adult SBWC goats (all over one year of age) were randomly selected from the same farm in Yulin, Shaanxi, China.”
Please clarify this aspect.
Line 128-129 and Figure 1 (Response 16): considering the response try to evaluate if deleting the figure. It is just a suggestion and not a request.
Line 31, I suggest to replace “polymorphism information content” with “polymorphic information content”.
Line 79, please use the same style in all the manuscript long to write 1109 (sometimes 1109 and sometimes 1,109).
Lines 95-98: even if, thanks to your explanations is now more understandable, I think it could be better to clarify that the authors extracted the DNA from all the 1109 samples and they carried out a preliminary screening using 50 samples in each pool. Soon after all the samples in the polymorphic pools were individually analyzed (if I understood well).
Line 197: Typing mistake “have shown”
Author Response

(The authors gave the same response as above.)
